# Structure of *Geobacter* OmcZ filaments suggests extracellular cytochrome polymers evolved independently multiple times

**Fengbin Wang[1], Chi Ho Chan[2], Victor Suciu[3], Khawla Mustafa[4], Madeline Ammend[2], Dong Si[3], Allon I Hochbaum[4,5,6,7]\*, Edward H Egelman[1]\*, Daniel R Bond[2]\***

[1]Department of Biochemistry and Molecular Genetics, University of Virginia School of Medicine, Charlottesville, United States; [2]Department of Plant and Microbial Biology, and BioTechnology Institute, University of Minnesota, St. Paul, United States; [3]Division of Computing and Software Systems, University of Washington Bothell, Bothell, United States; [4]Department of Chemistry, University of California, Irvine, Irvine, United States; [5]Department of Materials Science and Engineering, University of California, Irvine, United States; [6]Department of Molecular Biology and Biochemistry, University of California, Irvine, United States; [7]Department of Chemical and Biomolecular Engineering, University of California, Irvine, United States

*For correspondence:
hochbaum@uci.edu (AIH);
egelman@virginia.edu (EHE);
dbond@umn.edu (DRB)

**Abstract** While early genetic and low-resolution structural observations suggested that extracellular conductive filaments on metal-reducing organisms such as *Geobacter* were composed of type IV pili, it has now been established that bacterial *c*-type cytochromes can polymerize to form extracellular filaments capable of long-range electron transport. Atomic structures exist for two such cytochrome filaments, formed from the hexaheme cytochrome OmcS and the tetraheme cytochrome OmcE. Due to the highly conserved heme packing within the central OmcS and OmcE cores, and shared pattern of heme coordination between subunits, it has been suggested that these polymers have a common origin. We have now used cryo-electron microscopy (cryo-EM) to determine the structure of a third extracellular filament, formed from the *Geobacter sulfurreducens* octaheme cytochrome, OmcZ. In contrast to the linear heme chains in OmcS and OmcE from the same organism, the packing of hemes, heme:heme angles, and between-subunit heme coordination is quite different in OmcZ. A branched heme arrangement within OmcZ leads to a highly surface exposed heme in every subunit, which may account for the formation of conductive biofilm networks, and explain the higher measured conductivity of OmcZ filaments. This new structural evidence suggests that conductive cytochrome polymers arose independently on more than one occasion from different ancestral multiheme proteins.

## Editor's evaluation

This manuscript reports the CryoEM structure of OmcZ cytochrome nanowires of Geobacter sulfurreducens, the third cytochrome nanowire of Geobacter to be structurally resolved. OmcZ differs structurally from these previously determined nanowire structures, showing a different heme chain configuration. Based on these and other differences the authors speculate about the evolutionary origin of these nanowires and the mechanism of long-range electron transport. This manuscript is an important contribution to the field of electron transfer and will be of interest to everyone working in electron transfer and filament formation and interested in their evolution.

## Introduction

While soluble electron acceptors easily diffuse to the cytoplasmic membrane to support most microbial respirations, bacteria must build conductive pathways out of the cell to respire using insoluble metals and conductive surfaces. This process, known as extracellular electron transfer (ET), enables global Fe(III) and Mn(IV) reduction (*Gralnick and Newman, 2007*; *Lovley et al., 2004*), methane production in anaerobic digestors (*Morita et al., 2011*; *Shrestha et al., 2013*), methane oxidation in ocean seeps (*Chadwick et al., 2019*), corrosion (*Tang et al., 2019*), and electricity generation in microbial electrochemical devices (*Wang and Ren, 2013*).

When model organisms from the genus *Geobacter* utilize conductive surfaces as electron acceptors, they establish conductive multicellular biofilms (*Chadwick et al., 2019*; *Yates et al., 2015*). Many components, including pili and polysaccharides, are essential for the formation of these biofilms (*Reguera et al., 2005*; *Reguera et al., 2007*; *Rollefson et al., 2011*), but the octaheme *c*-type cytochrome OmcZ is the only cytochrome out of over 80 multiheme proteins in the *Geobacter sulfurreducens* genome necessary for long-distance conductivity in electrode associated biofilms (*Nevin et al., 2009*; *Peng and Zhang, 2017*; *Richter et al., 2009*). Deletion of *omcZ* affects anodic ET to electrodes (*Nevin et al., 2009*; *Peng and Zhang, 2017*; *Richter et al., 2009*), especially at low redox potential, and also slows cathodic corrosion of $Fe^0$ (*Tang et al., 2019*). In contrast, OmcZ is not needed for the reduction of other extracellular metals, including Fe(III) and Mn(IV) oxides (*Aklujkar et al., 2013*).

The *omcZ* gene (GSU2076) is upregulated 400–800% in conductive electrode biofilms (*Franks et al., 2012*; *Nevin et al., 2009*), producing a 50 kDa periplasmic preprotein that is processed by the OzpA protease (GSU2075) to a 30 kDa octaheme form (*Inoue et al., 2010*; *Jiménez Otero et al., 2018*; *Kai et al., 2021*). Similar operons containing homologous *omcZ* and *ozpA* are widespread throughout the Bacterial and Archaeal domains (*Chadwick et al., 2022*). In *G. sulfurreducens,* large heat-stable OmcZ molecules with a wide heme reduction potential window (-420 mV to –60 mV vs. the standard hydrogen electrode [SHE]) are abundant in the extracellular matrix between cells (*Inoue et al., 2011*), which under atomic force microscopy (AFM) imaging appear as ~2.5 nm diameter conductive filaments (*Yalcin et al., 2020*).

OmcZ represents one of three multiheme cytochrome filaments identified to date, along with the proteins OmcS (*Wang et al., 2019*; *Yalcin and Malvankar, 2020*) and OmcE (*Wang et al., 2022*). Cryo-EM structures of *G. sulfurreducens* OmcS and OmcE, which are involved in ET to insoluble Fe(III) and Mn(IV), reveal a core of closely spaced *c*-type hemes, with the unique characteristic of histidine residues from the previous subunit coordinating a heme from the next. While the OmcS and OmcE cytochromes share no amino acid sequence or structural similarity and show different patterns of surface glycosylation, the heme molecules in these two nanowires can be superimposed, suggesting a shared evolutionary path (*Wang et al., 2022*). In conductive-probe AFM measurements of enriched filaments cast on gold surfaces, OmcZ can demonstrate up to 1000-fold higher conductivity than preparations primarily consisting of OmcS (*Yalcin et al., 2020*), indicating that internal packing or external accessibility of hemes may be uniquely suited for interfacing OmcZ with electrodes.

In this work, we describe the cryo-EM structure of *G. sulfurreducens* OmcZ, and show that it forms a multiheme cytochrome nanowire different from OmcS and OmcE in both protein fold and heme arrangement. The core of closely spaced hemes shares no similarity with previously reported nanowires, two heme pairs are oriented at unique angles from one another, and one heme diverges from the central chain to create a solvent-exposed site along the wire. OmcZ also lacks the intersubunit coordination shared by OmcS and OmcE, and has no evidence of glycosylation. These data show that OmcZ is a member of a new and widespread multiheme cytochrome nanowire family that arose independently from OmcS and OmcE.

## Results

### Cryo-EM of the OmcZ filament

To obtain OmcZ filaments, we grew a Δ*omcS* strain of *G. sulfurreducens* on graphite electrodes poised at +0.24 vs. SHE. Filaments were enriched via shearing, DNAse treatment and salt precipitation, similar to protocols used for OmcS and OmcE, except that higher pH buffers were found to increase the recovery of OmcZ (*Figure 1—figure supplement 1*). Using cryo-EM (*Figure 1a*), we determined the structure of these filaments. Unlike OmcS filaments (*Wang et al., 2019*), OmcZ filaments did

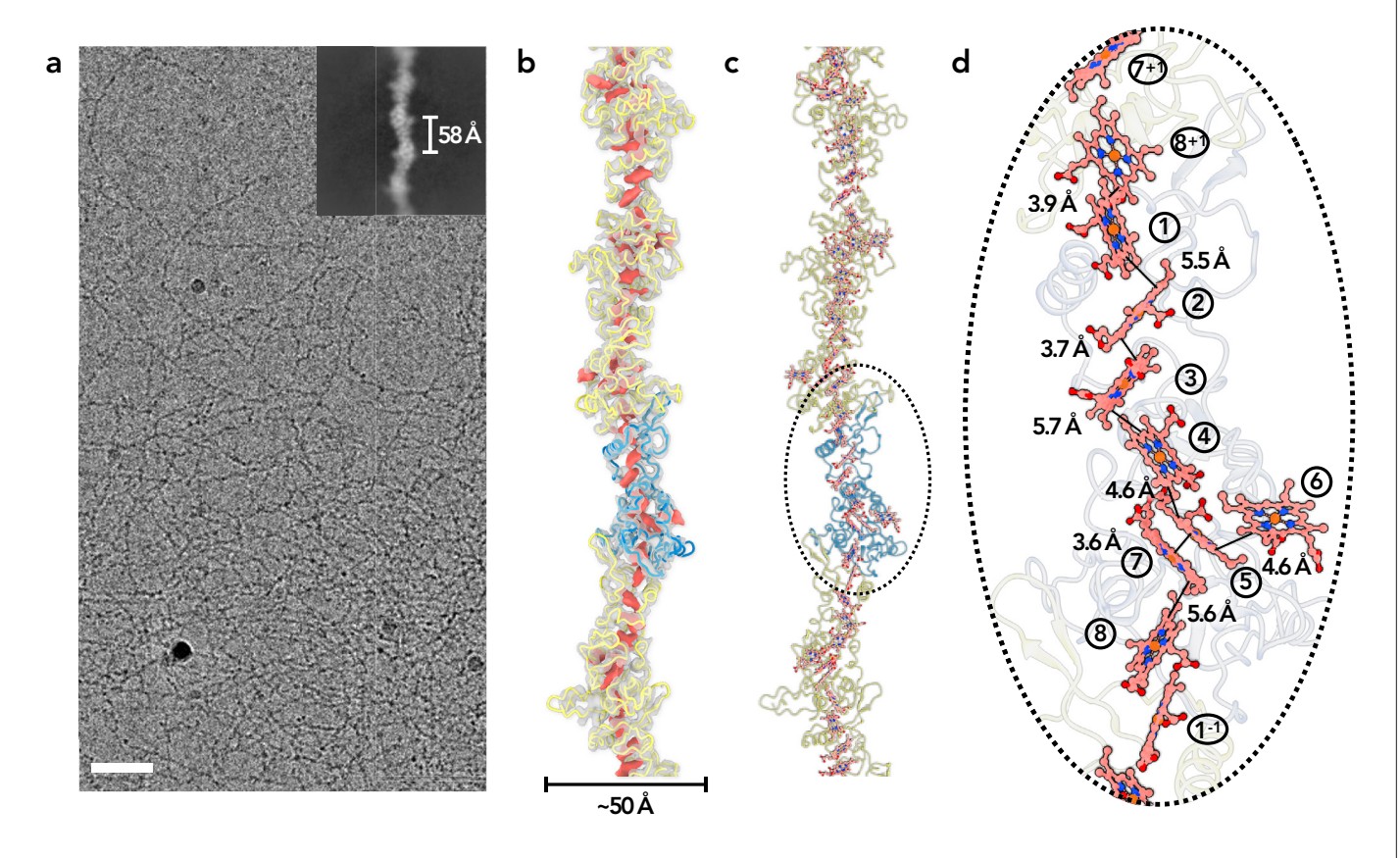

**Figure 1.** Cryo-EM of OmcZ filaments. (**a**) Cryo-EM image of the purified OmcZ filaments from Δ*omcS G. sulfurreducens* strain grown on electrodes. The sample was treated with DNase I prior to freezing. Scale bar = 50 nm. The upper right is a two-dimensional class average of the OmcZ filament, showing the rise of 58 Å between adjacent subunits. (**b**) The cryo-EM reconstruction (transparent gray) with backbone trace of the OmcZ subunits. The density of heme molecules is shown in red. One OmcZ subunit is colored in blue and the rest are yellow. (**c**) The atomic model of OmcZ filaments. The protein backbone trace is shown and the heme molecules are shown in ball and stick representation. (**d**) A zoomed region to show the heme array in OmcZ, with the minimum observed edge-to-edge distances indicated between adjacent porphyrin rings. Heme molecules are labeled with numbers in circles, The superscripts '+1' and '−1' indicate protein subunits above and below the central subunit, respectively.

The online version of this article includes the following source data and figure supplement(s) for figure 1:

**Figure supplement 1.** Isolation of OmcZ filaments.

**Figure supplement 1—source data 1.** Raw data for panels a, b, f and g.

**Figure supplement 2.** Cryo-EM analysis of the OmcZ filaments.

not show as strong a sinusoidal morphology (*Figure 1b and c*). An averaged power spectrum from raw segments (*Figure 1—figure supplement 2A*) showed a meridional layer-line at ~1/(58 Å), corresponding to the rise per subunit in the filament, and another layer-line near the equator at ~1/(132 Å) corresponding to the 132 Å pitch of a 1-start helix. After refining those parameters in the helical and subsequent non-uniform refinement, a ~4.2 Å resolution reconstruction was obtained, judged by a map:map Fourier shell correlation (FSC, *Figure 1—figure supplement 2b*). The model:map FSC gives a similar resolution estimation (*Table 1*). OmcZ filaments are ~50 Å at their widest point and coordinate eight heme molecules per cytochrome subunit.

Building an atomic model de novo at such resolution is typically challenging, especially when protein secondary structure elements are sparse (*Wang et al., 2022*). Fortunately, highly accurate protein structure prediction is now possible with AlphaFold2, even when the protein of interest has minimal similarity with any known protein structure (*Jumper et al., 2021*). The AlphaFold2 predicted full-length OmcZ model had a signal peptide, an N-terminal domain with eight pairs of histidines coordinating hemes that reasonably matched the cryo-EM map, and tandem Ig-domains (β-sandwiches)

**Table 1.** Cryo-EM and refinement statistics of OmcZ filaments.

| Parameter | OmcZ filament |
| --- | --- |
| **Data collection and processing** | |
| Voltage (kV) | 300 |
| Electron exposure (e$^-$ Å$^{-2}$) | 48 |
| Pixel size (Å) | 1.08 |
| Particle images (n) | 92,170 |
| Shift (pixel) | 60 |
| **Helical symmetry** | |
| Point group | C1 |
| Helical rise (Å) | 58.1 |
| Helical twist (°) | −158.2 |
| **Map resolution (Å)** | |
| Map:map Fourier shell correlation (FSC, 0.143) | 4.2 |
| Model:map FSC (0.5) | 4.4 |
| **Refinement and model validation** | |
| Ramachandran favored (%) | 88.0 |
| Ramachandran outliers (%) | 0.0 |
| Real space correlation coefficient | 0.80 |
| Clashscore | 22.1 |
| Bonds RMSD, length (Å) | 0.006 |
| Bonds RMSD, angles (°) | 0.990 |
| **Deposition ID** | |
| Protein Data Bank (model) | 8D9M |
| Electron Microscopy Data Bank (map) | EMD-27266 |

at the C-terminus (*Figure 1—figure supplement 2c*). Using the AlphaFold prediction as a starting model and iterative model refinements, residues P27 to S284 could fit into the map, with the signal peptide (residues 1–26) and the C-terminal Ig-domains missing. The RMSD between the cryo-EM model (255 Cα atoms) and the AlphaFold prediction is 5.9 Å, while the best aligned 113 atom pairs have an RMSD of 1.1 Å. It has been previously shown that OzpA, a subtilisin-like serine protease, cleaves the C-terminal part of OmcZ (*Kai et al., 2021*). While prior data detected amino acids consistent with residues 280–282 (amino acids FGN) at the C-terminus of purified OmcZ, the last visible C-terminal residues in the cryo-EM map extended to residue 284 (amino acids FGNSS) suggesting this could be the cleavage site. The next serine towards the C-terminus is S298, in the middle of a β-sandwich in the predicted model. The only other protein in *G. sulfurreducens* predicted by AlphaFold to have a similar structure is GSU1334, a putative octaheme c-type cytochrome. However, OmcZ was the only 30 kDa cytochrome detected by mass spectrometry of sodium-dodecyl sulfate-polyacrylamide gel electrophoresis (SDS-PAGE) separated filament preparations (*Figure 1—figure supplement 1*), consistent with transcriptional analysis showing OmcZ is induced to levels over 75-fold higher than GSU1334 during electrode growth (~50 RPKM for GSU1334 vs. 4200 RPKM for *omcZ*) (*Jiménez Otero et al., 2018*). An OmcZ model generated from homology modeling has been previously reported (*Yalcin et al., 2020*). However, neither its protein fold nor heme arrangements bear any resemblance to our experimentally determined model.

For all eight hemes per OmcZ subunit, two histidines axially coordinate iron at the center of each heme, and the vinyl groups of each heme form covalent thioether bonds with cysteines. Most heme-heme arrangements in OmcZ are either T-shaped or anti-parallel (*Figure 1d*). However, heme 6 (*Figure 1d*), in each OmcZ subunit does not fit into the closely packed central linear heme chain. The edge-to-edge distances between adjacent porphyrin rings in the main heme chain are between 3.6 Å and 5.7 Å, comparable to those in OmcS and OmcE filaments. All hemes in OmcZ are thus close to two other hemes, with the exception of heme 6 which is only close to one other heme, heme 5, at an edge-to-edge distance of 4.6 Å.

## OmcZ structurally differs from OmcS and OmcE

OmcZ filaments represent the third experimentally determined atomic structure of an extracellular cytochrome filament. Our first question was whether the protein fold or heme arrangement had been seen in other proteins. We have shown that OmcS and OmcE, while lacking sequence or protein fold similarity, share a conserved heme arrangement (*Wang et al., 2022*). In contrast, OmcZ shares no similarity to OmcS or OmcE in sequence, protein fold, or heme arrangement (*Figure 2a and b*). When we used the DALI (*Holm, 2020*) and Foldseek servers (*van Kempen et al., 2022*) to find structures with a similar fold to OmcZ, both servers returned no hits, suggesting the fold of OmcZ has not been

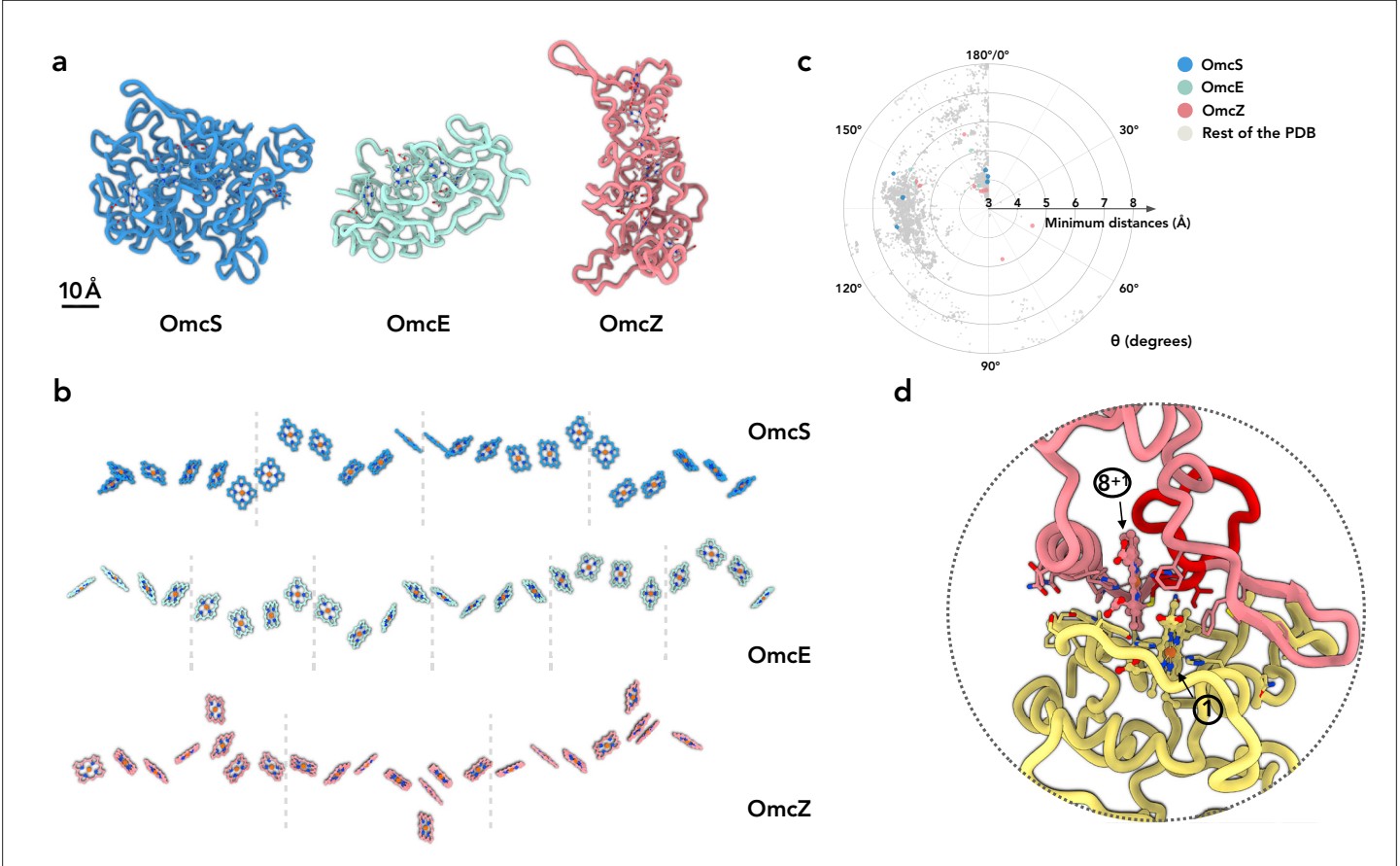

**Figure 2.** OmcZ differs from OmcS/OmcE in protein fold, heme arrangements and coordination. (**a**) The protein fold of OmcS, OmcE, and OmcZ. (**b**) The heme arrays of OmcS, OmcE, and OmcZ. Dashed lines indicate the separation of cytochrome protein subunits. (**c**) The heme-heme orientation plot. One heme can be aligned to the adjacent heme by a rotation and a translation. Only 25 non-hydrogen atoms in the porphyrin ring are used in the alignment. The heme (ID: HEC or HEM) pairs in all PDB structures were analyzed. The minimum distances refer to the smallest distance between two porphyrin rings, regardless of the atom type. The angle $\theta$ was determined from the alignment rotation matrix between heme pairs. For example, $\theta$ = 0° means two porphyrin rings are perfectly parallel, while $\theta$ = 180° means two porphyrin rings are perfectly antiparallel (flipped). All porphyrin rings pairs with a minimum distance less than or equal to 6 Å are shown. The porphyrin ring pairs in the OmcS, OmcE, and OmcZ filaments are highlighted in blue, green, and red, respectively. (**d**) The subunit-subunit interface in the OmcZ filament. The residues between C77 and H93 are highlighted in red. Heme molecules are labeled with numbers in circles.

previously observed. This may be due to the low percentage of secondary structure in OmcZ (18.2% helices and 4.7% β-strands), as the DALI and Foldseek servers also returned no hits for OmcS and OmcE, except for themselves and their homologs sharing extensive sequence similarity.

In the conserved heme packing of OmcS and OmcE (*Figure 2b*), the heme-heme interactions are periodic, with pairs of anti-parallel and T-shape repeating units. In OmcZ, one of eight hemes per repeating subunit (asymmetrical unit in the filaments) is located outside of the chain, so the minimum periodic pattern requires a unique doublet of anti-parallel hemes (anti-parallel, T-shape, anti-parallel, anti-parallel, T-shape, anti-parallel, T-shape). We previously analyzed preferred heme-heme orientations by looking at all 800+ heme *c* (HEC) containing proteins available in the Protein Data Bank (*Wang et al., 2022*). For comparison with OmcZ, we further improved the analysis by including other types of heme molecules (HEM, which is a non-heme *c*, in addition to HEC) and only including the atoms in the porphyrin ring for the analysis. Six out of eight heme pairs in OmcZ fall into the preferred clusters previously detected (*Figure 2c*). Interestingly, two heme pairs, heme 4-heme 5 and heme 5-heme 6 (*Figure 1d*), have a rare rotation angle of 56° and 82°, respectively. Heme pairs with a rotation angle between 50° and 90° normally have a 6 Å or larger edge-to-edge distances between porphyrin rings (*Figure 2c*). This is not the case in OmcZ: both of their edge-to-edge distances are smaller than 5 Å, suggesting efficient electron transport could occur.

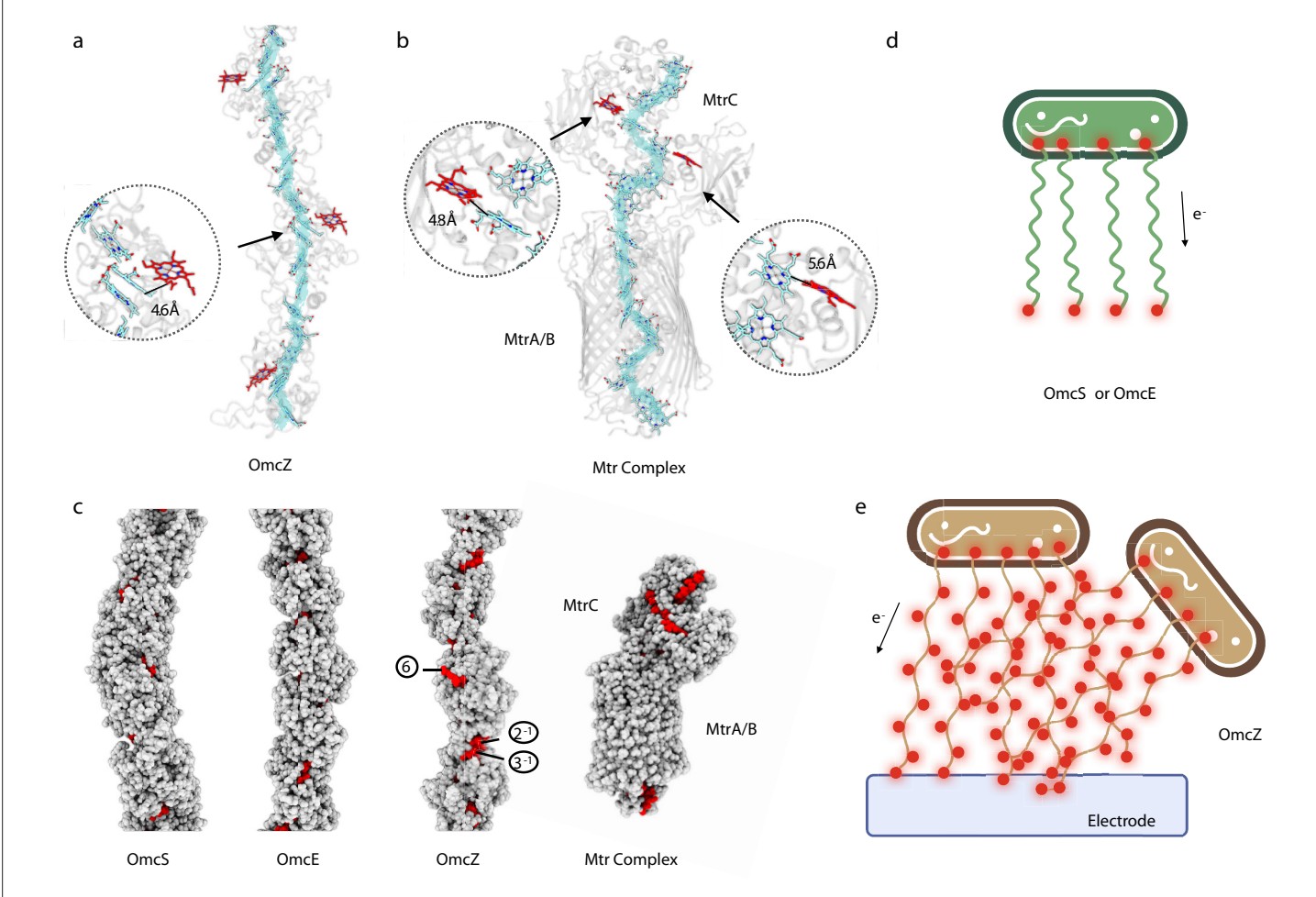

**Figure 3.** OmcZ filaments possess solvent-accessible hemes. (**a**) OmcZ and (**B**) Mtr complex (PDB 6R2Q). The main heme chain is traced in cyan while the solvent-accessible hemes are shown in red. The protein backbones are shown in transparent cartoon representation. The branched heme in OmcZ is 4.6 Å away from a heme in the linear chain, (**a**) while the two branched hemes in MtrC are 4.8 Å and 5.6 Å away from hemes in the linear chain (**b**). (**c**) Atomic models of OmcS filament, OmcE filament, OmcZ filament, and outer membrane-spanning protein complex MtrABC. Models without hydrogens are shown with atoms represented by spheres having the appropriate van der Waals radii. The protein residues are colored in gray while heme molecules are in red. The (**d**) and (**e**) schematic models of *Geobacter* cells producing different type of nanowires (OmcS or OmcE in (**d**) and OmcZ in (**e**)). Red dots indicate solvent-accessible heme molecules.

The protein-protein interface in OmcZ filaments is also quite different from OmcS and OmcE. In the latter cases, a heme molecule is coordinated by a histidine coming from an adjacent subunit. In OmcZ, all eight hemes are coordinated by histidines within the same subunit. The interfacial buried area in OmcZ is ~1200 Å$^2$, comparable to the interface in OmcE (~1100 Å$^2$) but much smaller than the interface in OmcS (~1900 Å$^2$). Similar to OmcS and OmcE, an anti-parallel heme pair is observed at the interface (*Figure 2d*). Also unique to OmcZ is the motif binding heme 1, which lies at the N-terminal interface. Instead of the canonical CxxCH motif, additional amino acids form a 14-residue loop between C77 and C92 (rather than the expected pattern of C89, C92, and H93). This additional loop is involved in the subunit-subunit interface (*Figure 2d*).

## OmcZ filaments possess solvent-accessible hemes

In OmcZ, heme 6 is largely solvent-exposed in every subunit in the filament (*Figure 3a*). In contrast, the only solvent-exposed hemes in the OmcE and OmcS polymers are those at the two ends of a filament, similar to the chain inside the 10 heme MtrAB membrane-spanning complex from Gram-negative *Shewanella baltica* (*Edwards et al., 2020*), where hemes are only solvent-exposed at the ends of a linear 'molecular wire' (*Figure 3b*). In contrast to the highly polymerized cytochromes in

OmcS, OmcE, and OmcZ, the MtrABC complex is a dimer of two 10-heme cytochromes, MtrA and MtrC, with MtrB acting as a surrounding insulator. It has been shown that the Mtr complex is capable of sustaining bidirectional electron transport rates greater than 8500 e s$^{-1}$ (*White et al., 2013*) and thus may provide some insights into the expected transport rates for the cytochrome filaments. The exposure of hemes in OmcZ was more analogous to the MtrC portion of the MtrABC complex, which has a branched heme chain introducing additional solvent-exposed hemes at the sides (*Figure 3b and c*). For comparison, the solvent-exposed area of heme 6 in OmcZ is 326 Å$^2$, while the most exposed heme in MtrABC (heme 901, the heme molecule exposed in the periplasm) has a solvent-exposed area of 292 Å$^2$. The other heme molecules in OmcZ are not comparable with heme 6 in terms of the solvent-exposed area, ranging from 12 Å$^2$ to 147 Å$^2$. As a control, the solvent-exposed area of heme molecules in OmcE ranges from 27 Å$^2$ to 85 Å$^2$; in OmcS they range from 52 Å$^2$ to 142 Å$^2$.

## Discussion

The observation that the *G. sulfurreducens* cytochromes OmcE and OmcS share the same highly conserved arrangement of hemes, even though the proteins themselves have no apparent sequence or structural similarity, was unexpected (*Wang et al., 2019*). These two proteins lack substantial amounts of secondary structure and mainly consist of loops and coils. The overall RMSD between OmcE and OmcS is 19 Å, which is what one might expect for two completely unrelated structures, but 28 atom pairs from these two proteins can be aligned with an RMSD of 1.1 Å. These 28 pairs represent the CxxCH heme-binding motifs and distal histidines that coordinate the hemes in both proteins. The four hemes in OmcE can therefore be superimposed almost perfectly on the first four of the six hemes in OmcS. This suggests that there is strong selective pressure on residues coordinating hemes in both proteins, but almost no selective pressure on the intervening residues that form loops and coils, allowing these two proteins to diverge from a common ancestor until the sequences, as well as overall folds, retain no recognizable similarity.

The structure of OmcZ reveals a filament that does not share the heme packing found in OmcS and OmcE. Furthermore, each of the eight hemes in an OmcZ subunit is coordinated axially by histidines from the same subunit, while in both OmcS and OmcE there is the coordination of hemes in an adjacent subunit by histidine in the neighboring subunit. This strongly suggests that polymers of multiheme cytochromes have arisen independently at least twice in bacterial evolution. Given the many known multiheme packing motifs (*Soares et al., 2022*), it is likely that other conductive cytochrome polymers exist.

Electron hopping is the likely mechanism of ET in OmcZ and other cytochrome polymers. Electron hopping is a charge transport process that links distinct, short-range ET steps, such as tunneling, into a long-range chain (*Warren et al., 2012*). In multiheme cytochromes, for example, an electron hopping pathway refers to a set of tunneling events between donor and acceptor states, such as closely coupled neighboring hemes, linked by slower ET processes. Even if the linking ET steps are also tunneling processes, the transport through a chain of many heme spanning distances greater than 30 nm, as is the case for cytochrome polymers, is accurately described as a series of discrete tunneling events (*Beratan et al., 2015*; *Ing et al., 2018*).

ET between adjacent hemes is slowed by increased distance and solvent exposure, and is more rapid between hemes in the T-shaped compared to parallel configuration (*Blumberger, 2015*; *Jiang et al., 2017*; *Jiang et al., 2020*; *van Wonderen et al., 2019*). In recent calculations, the solvent exposure of MtrC hemes combined with rate-limiting interfacial steps (up to 8 Å porphyrin-porphyrin spacing between hemes in MtrA and MtrC) was hypothesized to cause slower ET through MtrCAB compared to OmcS. While the spacing of porphyrin rings in OmcZ (3.6–5.7 Å) and OmcS/OmcE (3.4–6.1 Å and 3.8–6 Å, respectively) are similar, the increased solvent exposure of OmcZ and additional hemes in parallel configuration predict OmcZ could have a conductivity lower than OmcS. However, enriched OmcZ filaments are reported to have 1000-fold higher conductivity vs. OmcS (*Yalcin et al., 2020*).

The measured reduction potential values of heme in OmcZ span –420 mV to –60 mV (vs. SHE) (*Inoue et al., 2010*), similar to values obtained for OmcS (–360 mV to –40 mV vs. SHE) (*Qian et al., 2011*). To our knowledge, the reduction potential of OmcE has not been measured. Neither study determined whether these cytochromes were in their polymerized or monomeric state during reduction potential measurements but the reduction potential range of hemes in OmcS is comparable to

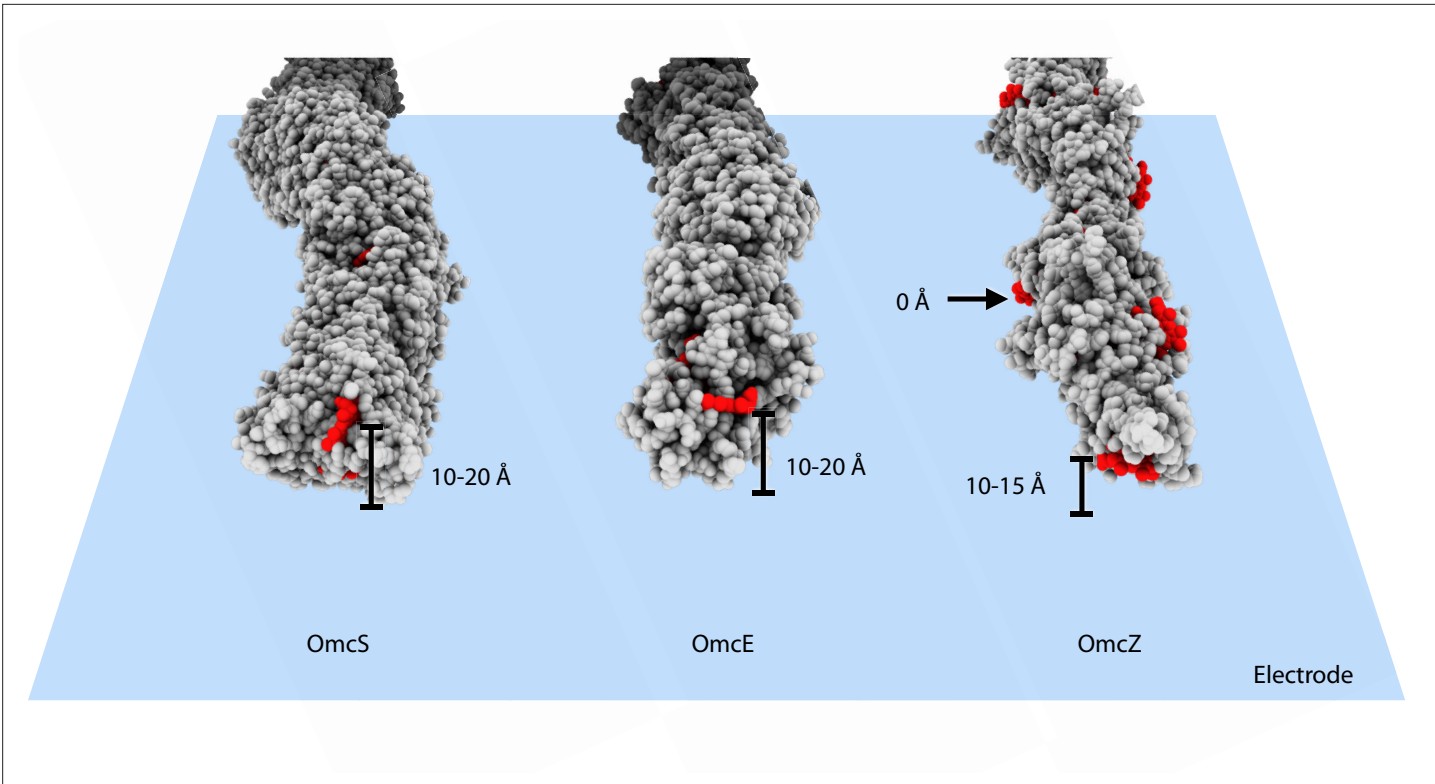

**Figure 4.** Heme arrangements may affect conductivity measurements. The atomic structures of OmcS, OmcE, and OmcZ filaments are shown. All protein atoms are colored in gray, while heme molecules are colored in red. When those filaments touch an electrode surface, shown as a blue plane, the estimated heme-to-surface distances are labeled. The possible tip proteins on these cytochrome filaments are not considered here.

computed values of heme reduction potential in OmcS polymers approximated as dimers (**Dahl et al., 2022**). The midpoint potential of these two cytochromes is also comparable, –220 mV and –212 mV vs. SHE for OmcZ and OmcS, respectively. The reduction potential ranges are consistent with those calculated of other multiheme cytochromes involved in extracellular ET processes with *bis*-His coordinated heme like in OmcZ and OmcS, such as *Shewanella oneidensis* MtrF (**Watanabe et al., 2017**) and MtrC (**Barrozo et al., 2018**). The direct determination of reduction potential from protein structure is non-trivial, requiring molecular dynamics and computational modeling. *Bis*-His coordinated hemes across all cytochromes in the Heme Protein Database have measured reduction potentials spanning –412 mV to +380 mV vs. SHE (**Reedy et al., 2008**). This broad range highlights the roles of the protein fold around the heme and the overall protein structure in determining reduction potentials. For OmcZ and OmcS (and OmcE), these folds are highly dissimilar (**Wang et al., 2022**), so speculating on the structural determinants of the reduction potential of the heme in OmcZ is premature at this time.

What could explain the higher apparent conductivity of OmcZ filaments (**Yalcin et al., 2020**)? First, measured conduction values are typically expressed as intrinsic conductivity, which involves an estimate of the cross-sectional area. This calculation likely distorts true values, as the heme chains provide the major contributions to electronic states supporting long-range conductivity, while the surrounding protein acts largely as an insulator (**Edwards et al., 2020**). For example, two 1 mm diameter copper wires with 0.5 mm and 2 mm thick insulation, respectively, would have the same measured conductance, but if scaled by cross-sectional area, the less insulated wire will have a calculated conductivity $(5/2)^2$ or 6.25 times more than the wire with thicker insulation. Similar errors would occur when comparing the larger diameter MtrABC complex or OmcS filament to the much thinner OmcZ. Consequently, length normalized conductance should be used when comparing heme polymer electronic transport properties. A second and likely greater difference could be introduced by differences in the proximity of heme to conductive surfaces, such as electrodes and AFM tips, used in measuring their electronic properties. The multiple exposed hemes of OmcZ could make very close contact with the electrode surface. In contrast, the exposed terminal heme in an OmcS or OmcE filaments may never

be closer than ~10 Å from the surface of an electrode or AFM probe, and due to glycosylation of their surfaces, OmcS and OmcE wires may be further insulated in conductance measurements (*Figure 4*).

The higher measured conductivity of OmcZ was proposed to be due to increased crystallinity and reduced disorder within OmcZ filaments compared to OmcS (*Yalcin et al., 2020*). However, the cryo-EM analysis reveals OmcZ filaments to be more flexible and less ordered than OmcS filaments. The earlier conclusion about crystallinity was based upon an increase in X-ray scattering from unoriented samples at ~1/(3.6 Å), which was interpreted to arise from parallel stacking of hemes. Such conclusions appear formally similar to an argument made previously (*Malvankar et al., 2015*) that the same increase in diffraction at ~1/(3.2 Å) in unoriented and dried samples was diagnostic of aromatic amino acid residue stacking in a hypothetical PilA-N structure.

Prior characterization of filaments assumed to be OmcZ (*Yalcin et al., 2020*) was based in large part upon scanning infrared nanospectroscopy, which found that at pH 7 putative OmcZ filaments contained ~41% of residues in β-sheets and ~39% of residues in α-helices, while at pH 2 they contained ~53% in β-sheets and ~20% in α-helices and in contrast, in the cryo-EM structure of filaments prepared at pH 10.5, OmcZ contained only ~18% helices and 5% β-sheets. These data suggest that IR nanospectroscopy is not yet a reliable method for determining filament identity or composition, just as measures of diameter, which have dominated the field of extracellular nanowires, are also unreliable (*Wang et al., 2022*).

With the availability of these new cytochrome filament structures, it is tempting to correlate their differences with phenotypes linked to each wire. The OmcE and OmcS extracellular wires of *Geobacter* are compatible with a model where filaments link a cell to a nearby metal oxide particle (*Figure 3d*). Post-translational modifications, especially the extensive glycosylations of OmcE, would be expected to add insulation to limit electron escape to the filament sides. In contrast, the OmcZ solvent-exposed heme raises the possibility that networks of OmcZ filaments may form with a multiplicity of paths through which electrons can flow, supporting a more conductive biofilm (*Figure 3e*). In fact, we observed by cryo-EM extensive meshes of OmcZ filaments (*Figure 1a*) that were not seen for OmcS and OmcE. Such junctions, whether they be the crossing over of two filaments or one filament forming a branch with another, could be paths for electron flow, while exposed hemes and lack of glycosylation along the surface of the network could offer multiple sites for ET to surfaces. Given the multitude of uncharacterized cytochromes in genomes of metal-reducing organisms and evidence that conductive filaments have arisen multiple times, this simple model will likely become increasingly complex as new structures become available.

## Materials and methods

### *G. sulfurreducens* growth and OmcZ filament preparation

*G. sulfurreducens* were grown in an anoxic basal medium with acetate (20 mM) as the electron donor and fumarate (40 mM) as the electron acceptor with 0.38 g/L potassium chloride, 0.2 g/L ammonium chloride, 0.069 g/L monosodium phosphate, 0.04 g/L calcium chloride dihydrate, 0.2 g/L magnesium sulfate heptahydrate and 10 mL mineral mix (*Chan et al., 2015*). The pH of the medium was adjusted to 6.8, buffered with 2 g/L sodium bicarbonate, and sparged with $N_2$:$CO_2$ gas (80:20) passed over a heated copper column to remove trace oxygen. For all experiments, *G. sulfurreducens* strains were revived anaerobically from frozen dimethyl sulfoxide stocks for single colony isolates on 1.2% agar plates. All cultures were grown at 30°C.

OmcZ filaments were isolated from electrode-grown *G. sulfurreducens* PCA Δ*omcS* strain (*Wang et al., 2022*). Three-electrode bioreactors with a working volume of 100 mL with 40 mM acetate as the electron donor were assembled as previously described (*Marsili et al., 2008*). The potential of two polished graphite working electrodes with a surface area of 9 cm$^2$ each was maintained at 0.240 V vs. SHE using a VMP3 multichannel potentiostat (Biologic), a stainless-steel mesh to serve as a counter-electrode and calomel reference. Transcripts of *omcZ* were ~6–10 times more abundant when electrodes were used as terminal electron acceptor compared to fumarate (*Jiménez Otero et al., 2018*). Reactors were inoculated 1:100 with fumarate-grown cells. Additional acetate (40 mM) was added to the reactor when the current reached ~0.5 mA/cm$^2$. Bioreactors were maintained at 30°C under a constant stream of humidified $N_2$:$CO_2$ (80:20) scrubbed free of oxygen by passage over a heated

copper furnace. In prior work, the oxygen concentration in the headspace of these reactors has been shown to be ~1 ppm.

   *G. sulfurreducens* biofilms were harvested when the current density reached ~1 mA/cm². Biofilm from eight 9 cm² electrodes was scraped into 80 ml of 150 mM ethanolamine buffer pH 10.5 (*Wang et al., 2017*) with 0.25 U/mL benzonase, as the higher pH buffers were found to increase recovery compared to pH 6.5 buffers used for OmcS and OmcE (*Figure 1—figure supplement 1*). OmcZ filaments were sheared using a Waring Commercial blender (Cat. No. 7011 S) for 3 min on low setting, then 1 min on high setting. Cell debris was removed by centrifugation at 8000 × *g* for 20 min.

## SDS-PAGE of filaments sample

The sheared filaments mixture was filtered and concentrated using 100 kDa and 300 kDa concentrator units (Sartorius Vivaspin) after multiple washing with buffer and subsequently ultrapure water. The concentrated filaments sample was diluted to 0.3 mg/mL with 50 mM ammonium bicarbonate (pH 7.8) buffer. The filaments sample was analyzed by SDS-PAGE to assess the separation as previously described (*Wang et al., 2019*). Briefly, Acryl/Bis (37.5:1, 40% w/v) solution (VWR) was used to prepare 4–16% Tricine-SDS-PAGE (*Schägger, 2006*) gels. Diluted filament samples with a final concentration of 0.15 mg/mL from each 100 kDa and 300 kDa filtration step were mixed with sample buffer with 0.5% and 2% w/v final concentrations of SDS. Before loading the samples into the gel, samples were boiled for 20 min before cooling to room temperature, followed by spinning at 2000 × *g* for 1 min. The samples were allowed to run with an initial voltage of 50 V for 10 min and the next voltage step of 200 V for 2.5 hr. A Spectra Multicolor Broad Range Protein Ladder (ThermoFisher) was used as mass standards for the protein bands. After washing the gels with ultrapure water three times, gels were stained with silver stain (*Kavran and Leahy, 2014*) or with 3,3′,5,5′-tetramethylbenzidine (*Thomas et al., 1976*).

## In-solution digestion by Trypsin/Chymotrypsin

In-solution digestion protocol was adapted from Promega protocols for Trypsin Gold and Chymotrypsin (Promega). A 0.3 mg/mL of filaments sample obtained after the 300 kDa filtration step was digested with a combination of Trypsin Gold and Chymotrypsin. For Trypsin Gold, 50 µg was mixed with 50 mM acetic acid to reach 1 µg/µL and diluted to 20 µg/mL in 50 mM ammonium bicarbonate (pH 7.8). For Chymotrypsin, 25 µg was resuspended in 1 mM HCl to a final concentration of 1 µg/µL and diluted to 20 µg/mL in 50 mM ammonium bicarbonate (pH 7.8). 5 µL of 100 mM DL-dithiothreitol (Sigma-Aldrich) was used to reduce 30 µL of the filaments sample followed by incubation at 37°C for 1 hr. The reduced sample was spiked with 10 µL of 100 mM iodoacetamide (Sigma-Aldrich) and incubated in the dark at room temperature for 45 min. Then, the mixture was diluted with 90 µL of 50 mM ammonium bicarbonate (pH 7.8) before adding 15 µL of the combination of Chymotrypsin and Trypsin Gold. The filamen sample was incubated for 3 hr at 37°C.

## Mass spectrometry of filament samples

The proteolyzed peptides mixture of filament sample was quenched after 3 hr incubation by adding 30 µL of 1% trifluoroacetic acid before MALDI-TOF/TOF (UltrafleXtreme, Bruker) characterization in positive ion mode. A saturated solution of α-cyano-4-hydroxycinnamic acid (Sigma-Aldrich) was used for the matrix solution by dissolving with a 2:1 solution of ultrapure water:acetonitrile (*Ing et al., 2017*). The sample spot was recrystallized on the MALDI plate with a 1:1 ratio of 0.1% of trifluoroacetic acid and acetonitrile. The data were analyzed using mMass protein identification software (*Niedermeyer and Strohalm, 2012*; *Strohalm et al., 2008*; *Strohalm et al., 2010*).

## Cryo-EM conditions and image processing

The cell appendage sample (ca. 3.5–4.0 µl) was applied to glow-discharged lacey carbon grids, and then plunge frozen using an EM GP Plunge Freezer (Leica). The cryo-EMs were collected on a 300 keV Titan Krios with a K3 camera (University of Virginia) at 1.08 Å/pixel and a total dose of ca. 48 e/Å². Motion corrections and contrast transfer function (CTF) estimations were done in cryoSPARC (*Punjani et al., 2017*; *Rohou and Grigorieff, 2015*; *Zheng et al., 2017*). Particles were auto-picked by 'Filament Tracer' with a shift of 60 pixels between adjacent boxes. All auto-picked particles underwent multiple rounds of 2D classification, and all particles in bad 2D class averages were removed. After this,

the OmcZ dataset had 92,170 particles remaining. The helical symmetry was originally determined in our previous study (*Wang et al., 2019*). After a helical refinement applying and optimizing the helical symmetry, a non-uniform refinement was performed to improve the resolution in the central area of the map. The resulting half maps were then sharpened by 'highres' mode in DeepEMhancer (*Sanchez-Garcia et al., 2021*), and the cryo-EM parameters are listed in *Table 1*.

## Model building of OmcZ filaments

The hand of the OmcZ filaments was determined by the hand of an α-helix. The hand assignment also agreed with the AlphaFold (*Jumper et al., 2021*) predictions of OmcZ. Part of the AlphaFold prediction (P27-S284), without N-terminal signal peptide and C-terminal β-sandwich domains, was docked into the cryo-EM map, and the regions that did not fit well were manually adjusted in Coot (*Emsley and Cowtan, 2004*). The de novo placement of heme molecules in the cryo-EM map at this resolution is challenging, especially since ligands are not included in the AlphaFold prediction. Therefore, to better refine heme-interacting areas at this resolution, bond/angle restraints for the heme molecule itself, His-Fe and Cys-heme thioester bonds were restricted based on the geometries obtained in high-resolution crystal structures such as NrfB20 (PDB 2P0B) and NrfHA21 (PDB 2J7A). Real-space refinement was performed with those restraints (*Afonine et al., 2018*). To clean up the protein geometry, OmcZ models were further rebuilt using Rosetta (*Wang et al., 2015*). MolProbity (*Williams et al., 2018*) was used to evaluate the quality of the filament model, and the refinement statistics are shown in *Table 1*.

## Structural analysis of heme c pairs

All structural coordinates with heme ligand (HEC or/and HEM) were downloaded from the Protein Data Bank. All possible heme pairs were then filtered with a minimum distance less than or equal to 6 Å, with the 'contact' command in UCSF-ChimeraX (*Pettersen et al., 2021*). For each qualified pair, the rotation matrix between two porphyrin rings was generated in ChimeraX using the 'align' command. The rotation angle θ was then calculated from the rotation matrix with the following equation, where tr is the trace of the rotation matrix:

$$|\theta| = \arccos\left(\frac{\mathrm{tr}(R)-1}{2}\right).$$

## Acknowledgements

The cryo-EM imaging was done at the Molecular Electron Microscopy Core Facility at the University of Virginia, which is supported by the School of Medicine and built with NIH grant G20-RR31199. This work was supported by NIH Grant GM122510 (E.H.E.), K99GM138756 (F.W.), DOE grant DE-SC0020322 (A.I.H., D.R.B., E.H.E., and M A.), AFOSR grant FA9550-19-1-0380 (A.I.H), NSF grant 2030381 (D.S.), Office of Naval research grant N00014-18-1-2632 (C.H.C), and the SRCP Seed Grant at the University of Washington Bothell (D.S).

## Additional information

### Competing interests

Edward H Egelman: Reviewing editor, *eLife*. The other authors declare that no competing interests exist.

### Funding

| Funder | Grant reference number | Author |
|---|---|---|
| National Institutes of Health | GM122510 | Edward H Egelman |
| National Institutes of Health | GM138756 | Fengbin Wang |

| Funder | Grant reference number | Author |
|---|---|---|
| Department of Energy | DE-SC0020322 | Edward H Egelman<br>Daniel R Bond<br>Allon I Hochbaum |
| Air Force Office of Scientific Research | FA9550-19-1-0380 | Allon I Hochbaum |
| National Science Foundation | 2030381 | Dong Si |
| Office of Naval Research | N00014-18-1-2632 | Chi Ho Chan |
| University of Washington SRP Seed grant | | Dong Si |

The funders had no role in study design, data collection and interpretation, or the decision to submit the work for publication.

## Author contributions

Fengbin Wang, Conceptualization, Validation, Investigation, Visualization, Writing – original draft, Writing – review and editing; Chi Ho Chan, Madeline Ammend, Investigation; Victor Suciu, Software, Formal analysis, Investigation; Khawla Mustafa, Resources, Investigation; Dong Si, Resources, Software, Formal analysis; Allon I Hochbaum, Daniel R Bond, Conceptualization, Resources, Formal analysis, Supervision, Funding acquisition, Writing – original draft, Writing – review and editing; Edward H Egelman, Conceptualization, Resources, Supervision, Funding acquisition, Writing – original draft, Project administration, Writing – review and editing

## Author ORCIDs

Fengbin Wang ⓘ http://orcid.org/0000-0003-1008-663X
Chi Ho Chan ⓘ http://orcid.org/0000-0002-6596-3436
Allon I Hochbaum ⓘ http://orcid.org/0000-0002-5377-8065
Edward H Egelman ⓘ http://orcid.org/0000-0003-4844-5212
Daniel R Bond ⓘ http://orcid.org/0000-0001-8083-7107

## Decision letter and Author response

Decision letter https://doi.org/10.7554/eLife.81551.sa1
Author response https://doi.org/10.7554/eLife.81551.sa2

# Additional files

## Supplementary files

• MDAR checklist

## Data availability

Atomic model deposited to the PDB with accession code 8D9M. Map deposited to the EMDB with accession code EMD-27266.

The following dataset was generated:

| Author(s) | Year | Dataset title | Dataset URL | Database and Identifier |
|---|---|---|---|---|
| Wang F, Chan CH, Mustafa K, Hochbaum AI, Bond DR, Egelman EH | 2022 | Cryo-EM of the OmcZ nanowires from Geobacter sulfurreducens | https://www.rcsb.org/structure/8D9M | RCSB Protein Data Bank, 8D9M |

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
