## [Editor Report]

This manuscript reports the CryoEM structure of OmcZ cytochrome nanowires of Geobacter sulfurreducens, the third cytochrome nanowire of Geobacter to be structurally resolved. OmcZ differs structurally from these previously determined nanowire structures, showing a different heme chain configuration. Based on these and other differences the authors speculate about the evolutionary origin of these nanowires and the mechanism of long-range electron transport. This manuscript is an important contribution to the field of electron transfer and will be of interest to everyone working in electron transfer and filament formation and interested in their evolution.

---

## [Decision Letter]

**Decision letter after peer review:**

Thank you for submitting your article "Structure of *Geobacter* OmcZ filaments suggests extracellular cytochrome polymers evolved independently multiple times" for consideration by *eLife*. Your article has been reviewed by 2 peer reviewers, and the evaluation has been overseen by a Reviewing Editor and Volker Dötsch as the Senior Editor. The following individual involved in review of your submission has agreed to reveal their identity: Marcus Joseph Edwards (Reviewer #1).

Essential revisions:

1) The structural model. With the emerging AlphaFold model prediction tool one should carefully explain how it was used for the non-expert readers. The authors should clearly refer to what was experimentally obtained, what was predicted by the AlphaFold model and the extension of the manual curation. This is important for the heme cofactors because manual curation can have a tremendous impact in the reported geometry of the hemes and consequent conclusions.

2) Related: It would be of general interest to give the rmsd of the final model and the initial alphafold model utilized in structure determination.

3) Solvent/exposed hemes. The branched heme is obviously an important structural target to putatively explain the higher conductivity of OmcZ compared to the other two filaments. However, I suggest that the authors compare the solvent exposure of all the hemes. I have the feeling that additional hemes are also more exposed in OmcZ.

4) Glycosylation surface. To the best of my knowledge OmcZ might have a sugar-binding domain. To clarify this point to the readers I suggest that the authors include an explanation on the OmcZ domains and discuss the possible location of the sugar-binding domain in the context of the preprotein and polymerized domain. See for example: AS Tauzin et al., (2016) mBio 7, e02134-02115.

5) Physiological conditions. I'm concerned regarding the experimental conditions in which the filaments are observed. The protocols matched those "used for OmcE and OmcS, except that higher pH buffer were used". The structural model is irrefutable, but a discussion on the experimental conditions (particularly pH 10.5) where filaments were observed versus the physiological bacterial growth conditions is missing. Is there any evidence for the formation of the filaments at physiological pH?

6) Electron hopping. Requires additional clarification for the non-expert readers.

7) Protein-protein interface stabilization in the OmcZ filament. Compared to OmcS and OmcE the authors state that "the protein-protein interface in OmcZ filaments is also quite different". The most striking observation is that in the case of OmcZ the adjacent subunits do not share axial histidine ligands. As written, it is implied that OmcZ is less stable, which is not clear to me. Besides the additional loop involved in the subunit-subunit interface, the authors should present further structural evidence that support the expected stability of OmcZ filament.

8) Redox potential values. The redox potential value for OmcZ is referred. The authors should add a structural-based discussion comparing the redox potential values of the three filaments and their putative functional differences.

*Reviewer #1 (Recommendations for the authors):*

The manuscript is well written, describing the work/results in a clear logical manner and providing the necessary context for the work.

I do have some suggestions which the authors may wish to consider implementing:

Results section

– "We have shown that OmcS and OmcE, while lacking sequence or structural similarity, share a conserved heme arrangement (Wang et al., 2022b)" – consider replacing "structural similarity" with "protein fold similarity" to further distinguish between the protein and hemes.

– "Asymmetrical subunit" – I found this terminology potentially confusing, consider replacing with "monomer" or "repeating subunit".

– It might be of general interest to give the rmsd of the final model and the initial alphafold model utilized in structure determination.

– When describing the Cx14CH motif figure 2D is referred to but the part of the structure/protein this is referring to is not clear from the figure.

– For clarity it might be useful to clarify that the heme 901 is heme 1 of MtrA which is exposed in the periplasm and is a proposed site for electron exchange.

Discussion section

– To prevent confusion avoid referring to CXXC by itself as a heme binding motif as a suitable proximal ligand (typically His) is also required. CXXC alone has been observed to form a disulfide bond in another extracellular multiheme cytochrome (OmcA).

Figures

Figure 1 – 1B/C – add a scale bar. 1D – state what numbers in circles represent.

Figure 2 – 2A/B – consider adding scale bars to both panels. 2B – Add more detail, were these aligned in some way? Say what the dashed lines represent. 2D – clarify the numbers in the circles in the figure legend.

Figure 4 – Provide a more descriptive legend.

*Reviewer #2 (Recommendations for the authors):*

From my reading the authors had not enough time to digest their impressive and new structural model. I understand that it is important to show for the first time an unpreceded structure, but I have no doubt that the outstanding quality of the research groups involved in this work can deliver to the readers an even sounder manuscript.

I am listing several main concerns/recommendations regarding the presented work.

1. The structural model. With the emerging AlphaFold model prediction tool carefulness is necessary for the non-expert readers. The authors should clearly refer to what was experimentally obtained, what was predicted by the AlphaFold model and the extension of the manual curation. This is important for the heme cofactors because manual curation can have a tremendous impact in the reported geometry of the hemes and consequent conclusions.

2. Solvent/exposed hemes. The branched heme is obviously an important structural target to putatively explain the higher conductivity of OmcZ compared to the other two filaments. However, I suggest that the authors compare the solvent exposure of all the hemes. I have the feeling that additional hemes are also more exposed in OmcZ.

3. Glycosylation surface. To the best of my knowledge OmcZ has a sugar-binding domain. To clarify this point to the readers I suggest the author include an explanation on the OmcZ domains and describe the location of the sugar-binding domain in the context of the preprotein and polymerized domain.

4. Physiological conditions. I'm very concern regarding the experimental conditions in which the filaments are observed. The protocols matched those "used for OmcE and OmcS, except that higher pH buffer were used". The structural model is irrefutable, but a discussion on the experimental conditions (particularly pH 10.5) where filaments were observed versus the physiological bacterial growth conditions is missing. Is there any evidence for the formation of the filaments at physiological pH?

5. Electron hopping. Requires additional clarification.

6. Protein-protein interface stabilization in the OmcZ filament. Compared to OmcS and OmcE the authors refer that "the protein-protein interface in OmcZ filaments is also quite different". The most striking observation is that in the case of OmcZ the adjacent subunits do not share axial histidine ligands. As written, it is implied that OmcZ is less stable, which is not clear to me. Besides the additional loop involved in the subunit-subunit interface, the authors should present further structural evidence that support the expected stability of OmcZ filament.

7. Redox potential values. The redox potential value for OmcZ is referred. The authors should add a structural-based discussion comparing the redox potential values of the three filaments and their putative functional differences.

---

## [Author Response]

Essential revisions:1) The structural model. With the emerging AlphaFold model prediction tool one should carefully explain how it was used for the non-expert readers. The authors should clearly refer to what was experimentally obtained, what was predicted by the AlphaFold model and the extension of the manual curation. This is important for the heme cofactors because manual curation can have a tremendous impact in the reported geometry of the hemes and consequent conclusions.

We agree with the reviewer on this matter; details have been expanded in the Methods. Briefly, the cryo-EM reconstruction was done independently without AlphaFold or prior knowledge. Then, AlphaFold prediction was used as a starting model and refined against the cryo-EM map. Ligands are not modeled in an AlphaFold prediction, and manual docking of the ligand into a map at ~4 Å resolution is often problematic. Therefore, we have always placed the hemes based on the c-type cytochrome geometries obtained in high-resolution crystal structures (including previously published OmcE and OmcS structures). Those geometric restraints include how the heme is orientated with respect to the CxxCH motif, the bond distance between histidine and Fe atom, the thioether bonds length between cysteines and heme, etc.

2) Related: It would be of general interest to give the rmsd of the final model and the initial alphafold model utilized in structure determination.

The RMSD between 113 pruned atom pairs is 1.1 Ångstroms (the part that the cryo-EM model best aligns with the AlphaFold prediction). If all 255 Cα presents in the cryo-EM model are used, the RMSD between the model and prediction is 5.9 Å. These numbers have now been included in the main text.

3) Solvent/exposed hemes. The branched heme is obviously an important structural target to putatively explain the higher conductivity of OmcZ compared to the other two filaments. However, I suggest that the authors compare the solvent exposure of all the hemes. I have the feeling that additional hemes are also more exposed in OmcZ

We thank the reviewer for this excellent suggestion. It turns out that only one heme molecule, heme 6 in OmcZ, is significantly more exposed to solvent than the other seven heme molecules. The total surface area of a heme molecule is about 815 Å2. The solvent accessible area for heme 1 to heme 8 in OmcZ are 17, 71, 147, 74, 12, 326, 77, and 40 Å2, respectively. As controls, the average solvent accessible area for hemes in OmcE and OmcS filaments are 53 (ranging from 27-85) and 96 Å2 (ranging from 52-142), respectively. This has now been added to the main text.

4) Glycosylation surface. To the best of my knowledge OmcZ might have a sugar-binding domain. To clarify this point to the readers I suggest that the authors include an explanation on the OmcZ domains and discuss the possible location of the sugar-binding domain in the context of the preprotein and polymerized domain. See for example: AS Tauzin et al. (2016) mBio 7, e02134-02115.

Based upon the AlphaFold prediction (that we take seriously) OmcZ has tandem Ig-domains at the C-terminus that are not present in the polymerized protein. We see no evidence that these might form a sugar-binding domain as in the Tauzin et al. paper. In fact, a large number of proteins (such as MALT1 and the muscle proteins myotilin, titin and filamin) all have tandem Ig-domains and there have been no suggestions that these are involved in sugar-binding. As for a glycosylation surface, that does not require a sugar-binding domain but requires specific residues that can be found in any domain for N-linked or O-linked glycosylation. We had previously described such a glycosylation surface in Wang et al., “An extensively glycosylated archaeal pilus survives extreme conditions.” Nature Microbiology 4, 1401-1410 (2019). In this pilus, one-third of the residues in a single Ig-domain were either serine or threonine, making them targets for O-linked glycosylation. We have now added a sentence discussing these tandem Ig-domains that are absent in the filament.

5) Physiological conditions. I'm concerned regarding the experimental conditions in which the filaments are observed. The protocols matched those "used for OmcE and OmcS, except that higher pH buffer were used". The structural model is irrefutable, but a discussion on the experimental conditions (particularly pH 10.5) where filaments were observed versus the physiological bacterial growth conditions is missing. Is there any evidence for the formation of the filaments at physiological pH?

All cells were grown under similar conditions, only the buffer used when shearing cell pellets was altered. This has been clarified in the text and two figures showing the evidence for better recovery in high pH buffers are added in Figure 1—figure supplement 1.

6) Electron hopping. Requires additional clarification for the non-expert readers.

We agree. We have added the following text to clarify the meaning of electron hopping:

“Electron hopping is the likely mechanism of electron transfer in OmcZ and other cytochrome polymers. Electron hopping is a charge transport process which links distinct, short-range electron transfer (ET) steps, such as tunneling, into a long-range chain (Warren et al. 2012). In multiheme cytochromes, for example, an electron hopping pathway refers to a set of tunneling events between donor and acceptor states, such as closely coupled neighboring hemes, linked by slower ET processes. Even if the linking ET steps are also tunneling processes, the transport through a chain of many heme spanning distances greater than 30 nm, as is the case for cytochrome polymers, is accurately described as a series of discrete tunneling events (Beratan et al. 2015, Ing et al. 2018).”

7) Protein-protein interface stabilization in the OmcZ filament. Compared to OmcS and OmcE the authors state that "the protein-protein interface in OmcZ filaments is also quite different". The most striking observation is that in the case of OmcZ the adjacent subunits do not share axial histidine ligands. As written, it is implied that OmcZ is less stable, which is not clear to me. Besides the additional loop involved in the subunit-subunit interface, the authors should present further structural evidence that support the expected stability of OmcZ filament.

It is very difficult (or nearly impossible at this point) to predict the stability of a protein polymer from a structural analysis of the subunit-subunit interface in the filament. The simplest measure, buried surface area, goes back to the classical work of Chothia and Janin, with the assumption that the stability is simply and directly related to the tightly bound water that would be liberated after polymerization. In this view, the loss of entropy of the protein is more than compensated for by the gain in entropy of the water. We do provide numbers for the buried surface area, and show that it is similar to OmcE but less than OmcS. We should not have suggested that OmcZ might be less stable as a filament than either OmcE or OmcS, and have clarified this.

8) Redox potential values. The redox potential value for OmcZ is referred. The authors should add a structural-based discussion comparing the redox potential values of the three filaments and their putative functional differences.

We have added the following text:

“The measured reduction potential values of heme in OmcZ span -420 to -60 mV (vs. standard hydrogen electrode, SHE) (Inoue et al. 2010), similar to values obtained for OmcS (-360 to -40 mV vs. SHE) (Qian et al. 2011). To our knowledge, the reduction potential of OmcE has not been measured. Neither study determined whether these cytochromes were in their polymerized or monomeric state during reduction potential measurements, but the reduction potential range of hemes in OmcS is comparable to computed values of heme reduction potential in OmcS polymers approximated as dimers (Dahl et al. 2022). The midpoint potential of these two cytochromes is also comparable, -220 and -212 mV vs SHE for OmcZ and OmcS, respectively. The reduction potential ranges are consistent with those calculated of other multiheme cytochromes involved in extracellular electron transfer processes with *bis*-His coordinated heme like in OmcZ and OmcS, such as *Shewanella oneidensis* MtrF (Watanabe et al. 2017) and MtrC (Barrozo et al. 2018). The direct determination of reduction potential from protein structure is non-trivial, requiring molecular dynamics and computational modeling. *Bis*-His coordinated hemes across all cytochromes in the Heme Protein Database have measured reduction potentials spanning -412 to +380 mV vs. SHE (Reedy et al. 2008). This broad range highlights the roles of the protein fold around the heme and the overall protein structure in determining reduction potentials. For OmcZ and OmcS (and OmcE), these folds are highly dissimilar (Wang et al. 2022), so speculating on the structural determinants of reduction potential of the heme in OmcZ is premature at this time.”